# Imaging a memory trace over half a life-time in the medial temporal lobe reveals a time-limited role of CA3 neurons in retrieval

Vanessa Lux[1], Erika Atucha[1,2,3], Takashi Kitsukawa[4], Magdalena M Sauvage[1,2,3]*

[1]Functional Architecture of Memory unit, Mercator Research Group, Medical Faculty, Ruhr University Bochum, Bochum, Germany; [2]Functional Neuroplasticity Department, Otto von Guericke University, Magdeburg, Germany; [3]Functional Architecture of Memory Dpt, Leibniz-Institute for Neurobiology, Magdeburg, Germany; [4]Graduate School of Frontier Biosciences, Osaka University, Osaka, Japan

**Abstract** Whether retrieval still depends on the hippocampus as memories age or relies then on cortical areas remains a major controversy. Despite evidence for a functional segregation between CA1, CA3 and parahippocampal areas, their specific role within this frame is unclear. Especially, the contribution of CA3 is questionable as very remote memories might be too degraded to be used for pattern completion. To identify the specific role of these areas, we imaged brain activity in mice during retrieval of recent, early remote and very remote fear memories by detecting the immediate-early gene *Arc*. Investigating correlates of the memory trace over an extended period allowed us to report that, in contrast to CA1, CA3 is no longer recruited in very remote retrieval. Conversely, we showed that parahippocampal areas are then maximally engaged. These results suggest a shift from a greater contribution of the trisynaptic loop to the temporoammonic pathway for retrieval.

*For correspondence:
magdalena.sauvage@gmail.com

**Competing interests:** The authors declare that no competing interests exist.

## Introduction

While it is a consensus that the hippocampus is engaged for the retrieval of recent memory, its involvement in retrieving more remote memories is still controversial. Some studies showing temporally-graded retrograde amnesia following hippocampal damage in humans have provided evidence that memory becomes hippocampal-independent as it ages and is ultimately 'stored' in the cortex (*Alvarez and Squire, 1994*: "the system consolidation theory"). An alternative theory posits that the hippocampus is not only engaged in retrieving recent memories but that it also supports cortical areas in retrieving more remote memories (*Nadel and Moscovitch, 1997*: the 'multiple trace theory'). The hippocampal subfields CA1 and CA3 are functionally segregated and quite a few studies have investigated their role in recent memory, for example in the retrieval of contextual fear conditioning (*Tanaka et al., 2014*; *Wheeler et al., 2013*; *Remaud et al., 2014*; *Daumas et al., 2005*; *Rampon et al., 2000*; *McHugh and Tonegawa, 2009*; *Hunsaker et al., 2009*; *Goshen et al., 2011*; see for reviews: *Nakazawa et al., 2004*; *Kesner, 2007*). However, it is unclear whether they contribute to a similar extent to the retrieval of more remote memories. A recent lesion study in humans indicates that CA1 participates to the retrieval of early and very remote memories (over 30 years-old) in addition to its well-established role for recent memories (*Bartsch et al., 2011*). In contrast, the role of CA3 in the retrieval of remote memory has not been yet studied in humans and has rarely been investigated in animals for memories older than a month, a period of time roughly equivalent

**eLife digest** There are two schools of thought about what role the hippocampus – a region of the brain – plays in memory. Some neuroscientists think that it is involved in retrieving all memories. Others believe that its contribution is restricted to the retrieval of recent memories, while a neighboring part of the brain called the parahippocampal region takes over to retrieve older memories.

The hippocampus contains two distinct areas called CA1 and CA3, which have recently been suggested to have, at least partially, separate roles. For example. previous studies have shown that CA3 plays an important role in processes that tend to be less efficient as time goes by. However, it remains unclear whether CA1 and CA3 contribute equally to the retrieval of recent and older memories.

Lux et al. addressed this question by observing brain activity in mice as they retrieved recent and older memories. The experiments show that both areas of the hippocampus are involved in retrieving recent memories, but that only the CA1 area is involved in the retrieval of older memories. The parahippocampal region is much more active during the retrieval of older memories than recent ones.

These findings clarify the role of the hippocampus in memory by showing that it is involved in the retrieval of both recent and older memories. The next steps will be to better understand how the CA1 and CA3 areas contribute to memory and to pin point the specific molecular mechanisms these regions rely on to do so.

to four years in humans based on life expectancy. Hence, very little is known about the extent to which CA3 contributes to the retrieval of remote and very remote memory. Nonetheless, computational studies suggest that CA3 would contribute to retrieving memories, at least recent and early remote memory, via processes involving the completion of memory representations based on partial or degraded features of these representations (the 'pattern completion' theory: *Rolls et al., 1997*). However, the memory for single features and/or details that could serve as retrieval cues degrades over time (*Wiltgen and Silva, 2007*; *Wiltgen et al., 2010*). Hence, it raises the question whether CA3 would also contribute to the retrieval of very remote memories or whether its contribution depends on the age of the memory trace.

In addition, a critical involvement of the parahippocampal region in the retrieval of recent and early remote memory is supported by a handful of animal studies and by recent reports of flat retrograde amnesia in patients with damage extending from the hippocampus to other medial temporal lobe areas. However, these studies did not investigate very remote memory traces while the contribution of cortical areas to the retrieval of memory is thought to be maximal at this stage. (*Squire and Alvarez, 1995*; *Gusev et al., 2005*; *Bucci et al., 2002*; *Burwell et al., 2004*; *Izquierdo et al., 1997*). Moreover, despite evidence for a functional segregation between parahippocampal areas in terms of spatial/non-spatial information processing and/or of the mediation of distinct memory processes (*Mishkin et al., 1983*; *Eichenbaum et al., 2007*; *Beer et al., 2013*), no study has yet dissociated the specific contribution of the perirhinal (PER) and postrhinal cortices (POR) nor that of the lateral and medial entorhinal cortices (LEC, MEC) to the retrieval of remote memory.

Here, we address these issues using a contextual fear conditioning task by testing whether CA3 contributes to a comparable extent as CA1 to the retrieval of recent (1 day and 1 week-old), early remote (1 month-old) and very remote memories (6 months or 1 year-old), the latter corresponding roughly to 20 and 40 years-old memories in humans based on life expectancy which are among the delays the most investigated in studies focusing on very remote memory (*Figure 1A*). In addition, we investigated the extent to which the PER, the POR, the LEC and the MEC differentially contribute to the retrieval of these memories. Because performing in-vivo electrophysiology recording simultaneously in six distant brain areas remains a major challenge and using a lesion/inactivation approach in adjacent brain areas in mice was unlikely to yield the spatial resolution necessary to tease apart the specific function of each area, a high-resolution molecular imaging technique (e.g. to the cellular

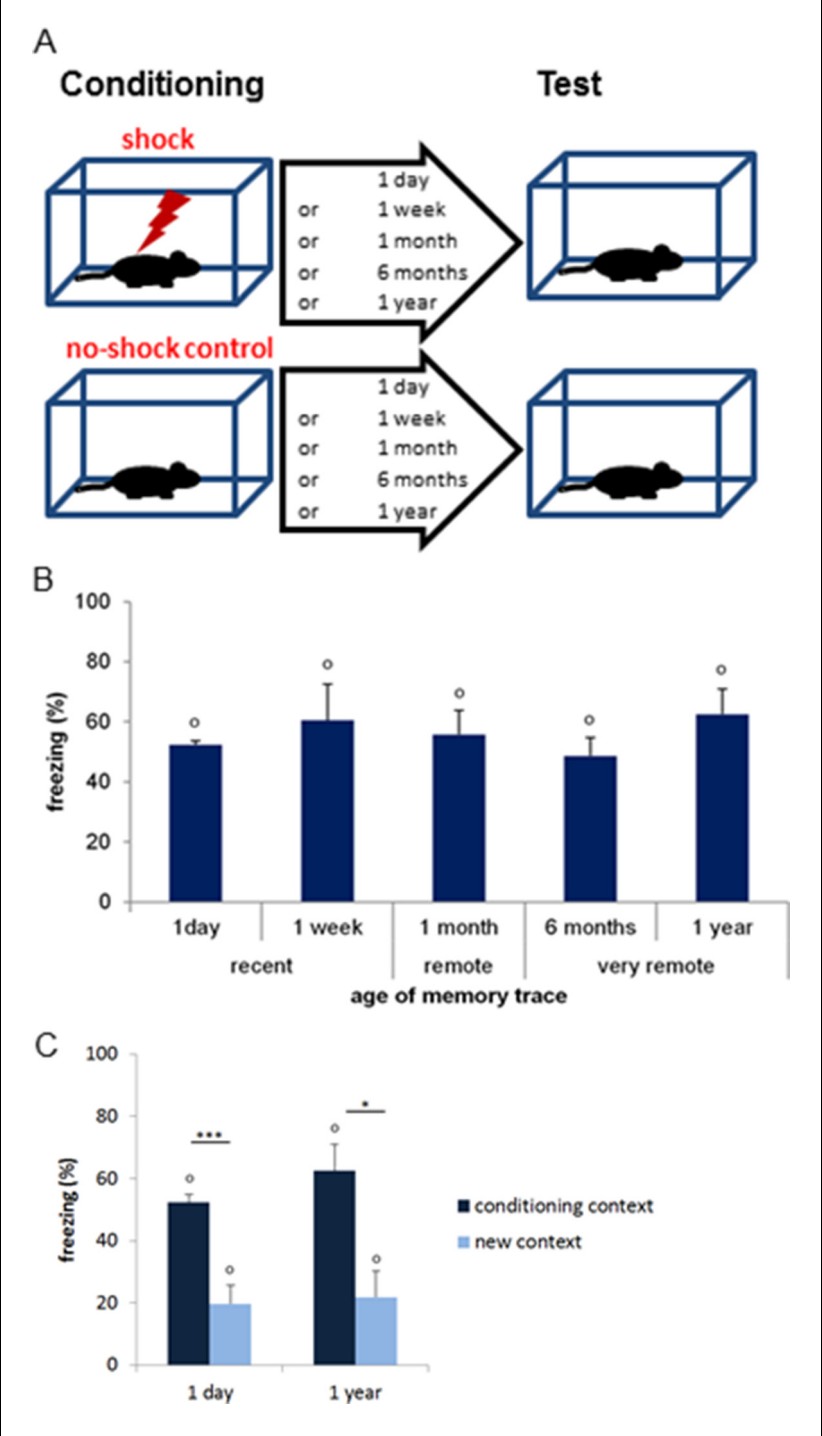

**Figure 1.** Contextual fear-conditioning task and memory performance. (**A**) Schematic description of the task. Mice were exposed to a conditioning chamber they explored freely for 2.5 min after which they received a single 1 mA footshock and were returned to their home-cage after 3.5 additional min. Memory for the association of the footshock with the conditioning context was tested after either 1 day, 1 week, 1 month, 6 months or a year by re-exposing the conditioned mice to the conditioning chamber and measuring freezing levels (n=4 mice per delay; n=20 total). Upon completion of the task, mice were sacrificed and their brain processed for imaging. 'No-shock' age-matched control groups were exposed to the exact same experimental conditions but did not receive any footshock (n=4 mice per delay; n=20 total). Of note, additional age-matched home-cage control groups were generated to control for *Arc* baseline expression (n=4 mice per delay; n=20 total; not shown here). (**B**) Differences in freezing indices between 'shock' and 'no-shock' groups (e.g. normalized freezing indices) in function of the age

*Figure 1 continued on next page*

*Figure 1 continued*

of the memory trace. All shocked groups showed stronger freezing than their no-shock controls at test as shown by differences in freezing levels significantly higher than 0 (a 55.9 ± 3.4% increase in average). Importantly, freezing levels were similar across delays, suggesting that memory strength did not significantly differ as memory aged. (C) Normalized freezing index induced by exposure either to the conditioning chamber or to a new context. Two additional age-matched groups ('new context' groups) were conditioned and normalized freezing levels were evaluated as mice were exposed at test to a new context instead of the conditioning chamber one day or one year after conditioning (n=4 mice per delay; n=8 total). Conditioned mice reexposed to the conditioning chamber froze in average 36 ± 4.7% more than the conditioned mice tested in the new context, demonstrating that the freezing levels were specific to the conditioning chamber. Error bars are mean ± SEM. 'o' indicate a significant comparison to 0 at p<0.05; asterisks a significant effect of the context at p<0.05 for '*' and p<0.01 for '***'.

level) was employed. This imaging technique is based on the detection of the expression of the immediate-early gene *Arc* which has been especially linked to plasticity processes and cognitive demands and has been recently used for mapping cognitive processes in the medial temporal lobe (*Guzowski et al., 1999*; *Nakamura et al., 2013*; *Sauvage et al., 2013*; *Figure 2B–G*).

## Results

### Memory performance is comparable across delays and freezing levels are specific to the conditioning context

Freezing levels upon re-exposure to the conditioning context were high and did not significantly differ between delays, indicating that the memory for the footshock/context association was strong and that its strength was comparable between delays whether the memory was recent or remote (comparisons to 0: all ps<0.015, in average a 55.9 ± 3.4% increase in freezing compared to age-matched 'no-shock' groups; F(4,15) =0.26, p=0.897; *Figure 1B*). Of note, statistical analysis of the data with or without normalization, e.g. direct analysis of the% resting time at test, yield the same results. Importantly, this result shows that a difference in memory strength across delays cannot account for the differences in the pattern of activation reported in the present study. In addition, we tested that the freezing behavior observed during re-exposure was specific to the conditioning context by exposing two additional conditioned groups to a new context either 1 day or 1 year after conditioning. In this case, freezing levels were strikingly lower than those observed by re-exposing the mice to the conditioning chamber, demonstrating that the freezing behavior was indeed specific to the conditioning context (one day: conditioning context: 52.4 ± 2.5% vs new context: 19.8 ± 5.9%; one year: conditioning context: 62.5 ± 8.4% vs new context: 21.9 ± 8.385%; both ts>3.9 and ps<0.007; *Figure 1C*).

Of note, a time-dependent generalization of the freezing behavior to a similar context has been reported in some studies testing subjects 15 to 42 days after conditioning i.e. for the retrieval of early remote memory (*Winocur et al., 2007*; *Wiltgen et al., 2010*; *Wiltgen and Silva, 2007*) as well as a generalization to different contexts depending on experimental conditions (multiple footshocks, preexposure to conditioning context etc..) (*Wang et al., 2009*; *Kitamura et al., 2012*; *Biedenkapp and Rudy, 2007*). No such generalization was seen in our study one year after conditioning (*Figure 1C*) as the new testing context was designed to be as different as possible from the conditioning context (in shape, color, cues, heights and width, odor, background noise, lightning and texture), the protocol did not include preexposure to the conditioning context and only one footshock was delivered. Similar findings were also observed in studies investigating the retrieval of memory at remote delays (50 days and 16 months, respectively; *Anagnostaras et al., 1999*; *Gale et al., 2004*).

### Supplementary information: analysis of the freezing levels with or without normalization (e.g. direct analysis of the% resting time at test) yields the same results

An alternative way to express conditioned fear responses is to report the% resting time at test. However, as mentioned in the material and methods, the conditioning baseline of one of the ten groups

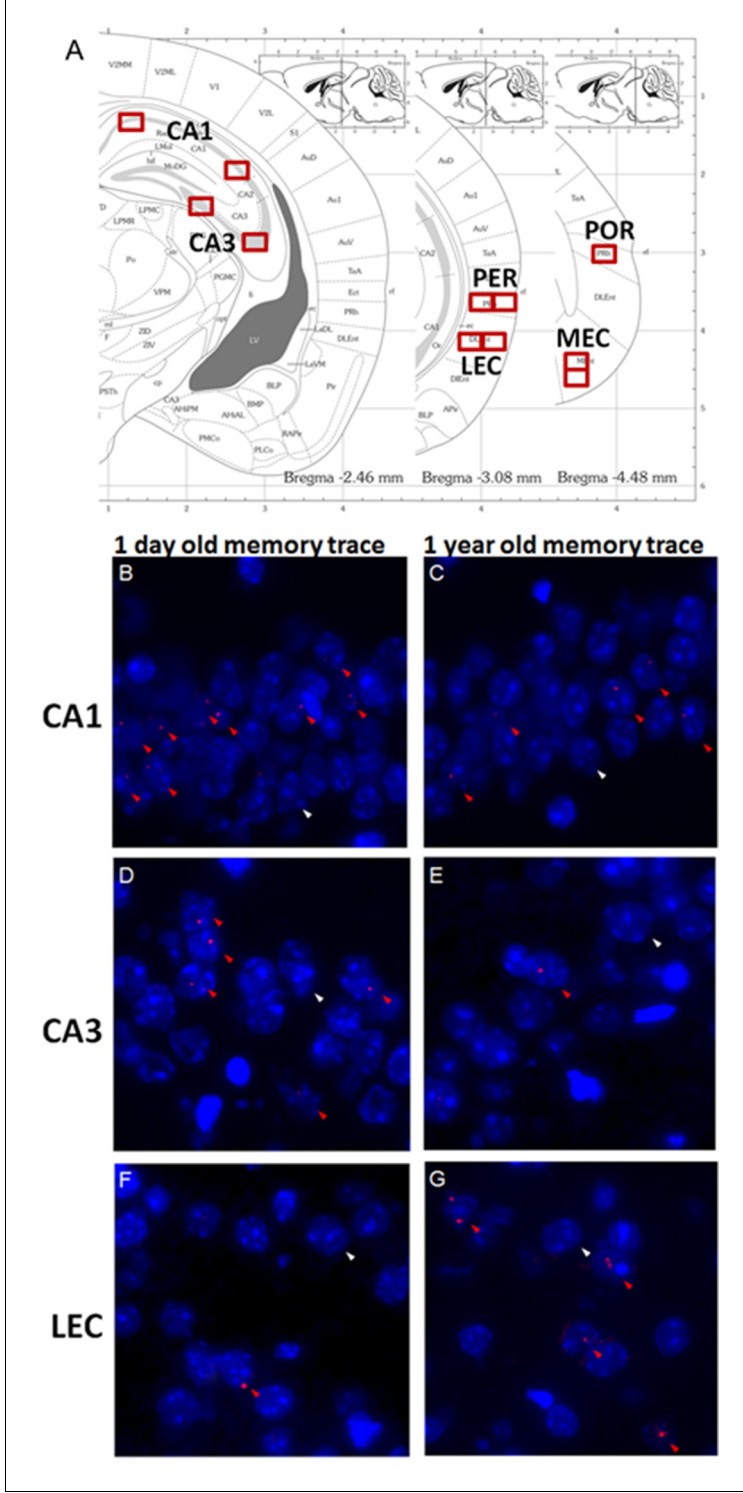

**Figure 2.** Imaging brain activity in the medial temporal lobe (MTL). (**A**) Location of counting frames for CA1, CA3, the perirhinal (PER), the lateral entorhinal (LEC), the postrhinal (POR) and the medial entorhinal cortices (MEC). Task-induced *Arc* nuclear RNAs' expression was detected on three nonconsecutive brain slices for each target area covering approx. 800 microns. (**B-G**) Cells activated during the retrieval of one day–old and one year-old memory traces in CA1, CA3 and the LEC: *Arc* positive cells (red arrowheads) and an exemplar of non-activated cells *Arc* negative cells (white arrowheads). Cell nuclei are labeled in blue with DAPI. (**B,C**) CA1 is engaged independently of the age of the memory trace while (**D,E**) activity levels in CA3 are negligible for very remote

*Figure 2 continued on next page*

*Figure 2 continued*

memories. (**F-G**) The parahippocampal areas, for example the LEC, are maximally engaged during the retrieval of very remote memories.

studied (the one year-shock group) was slightly lower than that of the other shock groups ($F_{(4,15)}$ =3.37, p=0.037; post-hoc Tukey: 1 year vs. 1 day p=0.041; vs. 1 month or 6 months: p=0.079 and p=0.071, respectively; importantly though, no significant shock vs no-shock effect was found at this delay: t(6)=1.97; p=0.131). Since a similar pattern was found at test (shock groups: $F_{(4,15)}$=6.85, p<0.005, post hoc Tukey: 1 year vs. all other delays ps<0.010, except 6 months p=0.068) focusing solely on the% resting time at test could have led to a misinterpretation of the test data since the lower freezing index at test is carried by a lower baseline at conditioning. Therefore, freezing levels discussed in the manuscript have been normalized by the conditioning baseline for each group. Nevertheless, for the sake of transparency, we also performed the analysis on non-normalized freezing levels (e.g. on% resting time at test). This analysis led to the same conclusion as the analysis of the normalized data: shocked animals display significantly higher freezing levels than the no-shock groups at all delays, reflecting a successful memory for the context-footshock association (two-way ANOVA with 'shock' and 'delay' as factors and '% resting time at test' as dependent variable: shock effect: $F_{(1,30)}$=275.9, p<0.001; delay effect: $F_{(4,30)}$=15.147, p<0.001; interaction shock x delay effect: $F_{(4,30)}$=1.21, p=0.328; all ts>5.25, ps<0.012).

## CA3's activation is limited to recent and early remote memories while CA1 is recruited independently of the age of the memory trace

First, we tested if CA3 and CA1 were differentially recruited as memory aged and especially if CA3 was engaged for the retrieval of very remote memories. Statistical analysis of the task-induced *Arc* expression revealed that, the contribution of CA3 to the retrieval of memory is restricted to that of recent and early remote memory traces (e.g. to up to 1 month-old memories) while it was negligible for very remote memories (6 months- and 1 year-old memory traces; *Figure 3A*). Indeed, CA3 was no longer engaged for memory retrieval at this time point ($F_{(4,15)}$=26.73, p<0.001; Tukey posthoc: recent or early remote traces vs very remote traces: all ps<0.028; comparisons to 0: up to 1 month-old traces: ps<0.010, 6 months and 1 year old traces: n.s). Importantly, this was only the case for CA3 as revealed by a significant region by delay interaction and further statistical analysis showing that, in contrast to CA3, CA1 was engaged for all delays and overall more activated than CA3 (CA1 and CA3 comparisons to 0: all ps<0.033; area effect: $F_{(1,30)}$=62.74, p<0.001; delay effect: $F_{(4,30)}$ =20.15, p<0.001; region by delay interaction effect: $F_{(4,30)}$=5.84, p=0.001; post-hoc analysis: CA1 vs CA3: all ts>3.89 and ps<0.030 but for 6 months t=2.36, p=0.099 and 1 week n.s; *Figure 3A*). In addition, within-area analysis showed that activation levels in CA1 were higher one day after conditioning compared to all other delays but comparable between the remaining delays ($F_{(4,15)}$ = 9.46, p=0.001; Tukey posthoc: one day higher than any other delay: all ps<0.041 but for 1 month p=0.090; one week to one year: all ps>0.083 but 1 month vs 6 months: p=0.038). Of note, analysis of the task-induced *Arc* expression with or without normalization yields the same results. This new finding of a time-limited involvement of CA3 for the retrieval of memory was further supported by significant correlations between memory performance and *Arc* levels in CA3 for recent and early remote memories, which failed to reach significant for very remote memory traces (recent and early remote: $R^2$=0.47, p<0.001; very remote: $R^2$=0.06, p=0.351; *Figure 3B–C*) while correlations were significant for CA1 independently to the age of the memory trace ($R^2$s.>0.40, ps<0.008; *Figure 3D, E*). The lack of involvement of CA3 in the retrieval of very remote memories was further strengthened by the fact that levels of activation in CA3 did not differ between the memory-impaired and the memory-intact animals that were tested one year after conditioning (intact vs impaired: freezing index: t(6) = 4.32, p=0.005, *Arc* expression: t(6)=1.08, p=0.321, comparison to 0: n.s; see *Figure 4A and B*). In a striking contrast and supporting a crucial role of CA1 in the retrieval of very remote memories, CA1 was not recruited in the memory-impaired group as opposed to the memory–intact group (intact vs impaired: t(6)=3.38, p=0.015; comparisons to 0: memory impaired: t(3)=0.08, p=0.936; memory intact: t(3)=6.69, p=0.007; see *Figure 4B*). Of note, these results are unlikely due to an age- related decline specific to CA3 since baseline *Arc* expression was comparable between

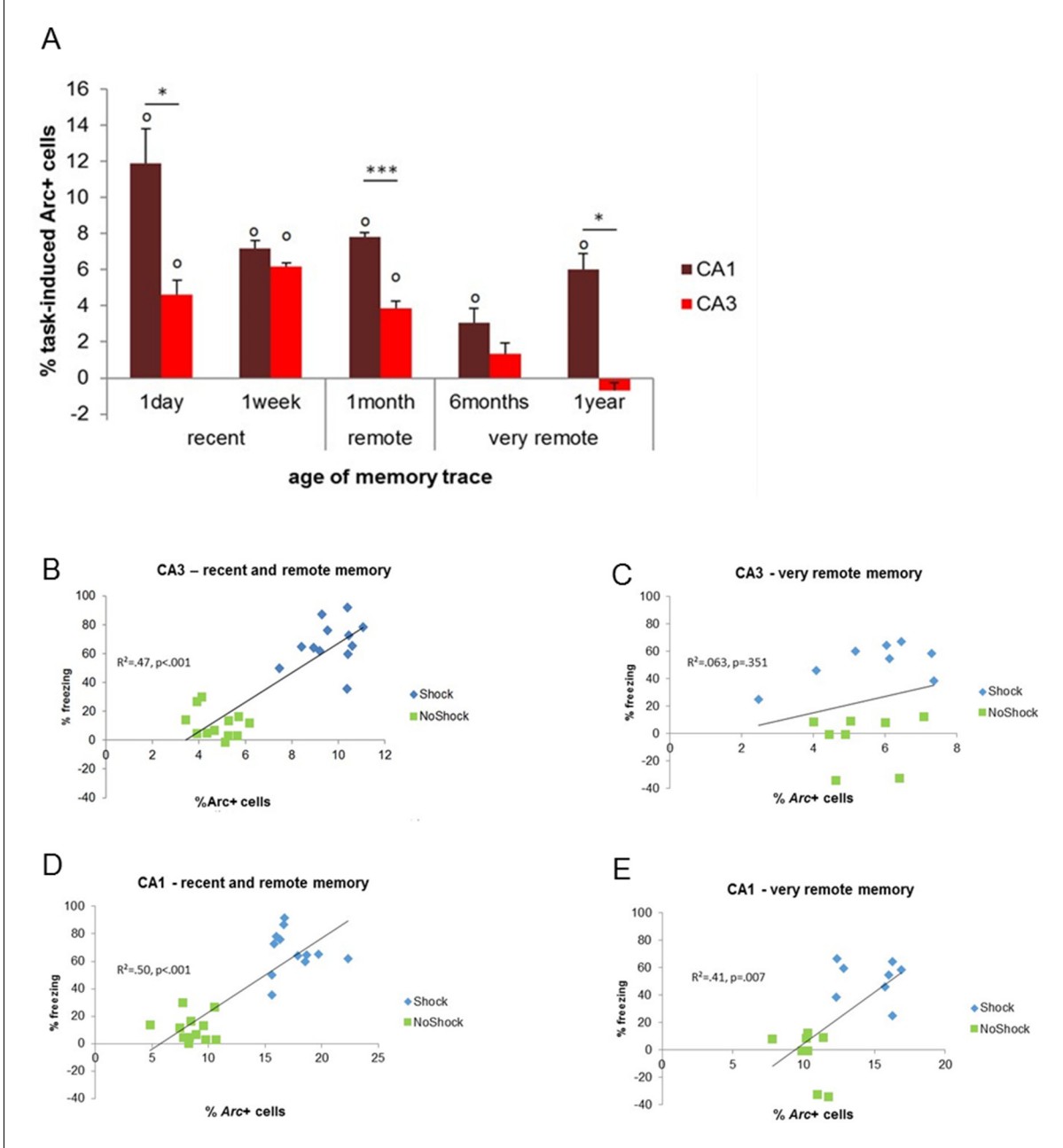

**Figure 3.** Activity patterns in CA1 and CA3 over time and correlations with memory performance. (**A**) CA3's contribution to memory retrieval depends on the age of the memory trace, not CA1's, as CA3's was no longer significantly recruited for the retrieval of very remote memories. (**B,C**) CA3's activity levels were predictive of memory performance only for the retrieval of recent and early remote memories while **D,E**) CA1's activity was independently of the age of the memory trace, Error bars are mean ± SEM. 'o' indicate a significant comparison to 0 at p<0.05; asterisks a t-test at p<0.05 for '*' and at p<0.01 for '***'.

and within CA1 and CA3 over time as shown by the absence of significant 'area' x 'delay' interaction, main area effects and posthocs analysis over time failing to reach significance (area effect: ($F_{(1,30)}$= 2.30; p=0.14), delay effect: $F_{(4,30)}$=0.77;p=0.55, area by delay interaction effect: ($F_{(4,30)}$=40; p=0.81; posthocs: delay effects: CA1: $F_{(4,20)}$= 0.63, p=0.65; CA3: $F_{(4,20)}$=0.52, p=0.72) and because *Arc* expression elicited by a maximal electroconvulsive shock, a standard positive control

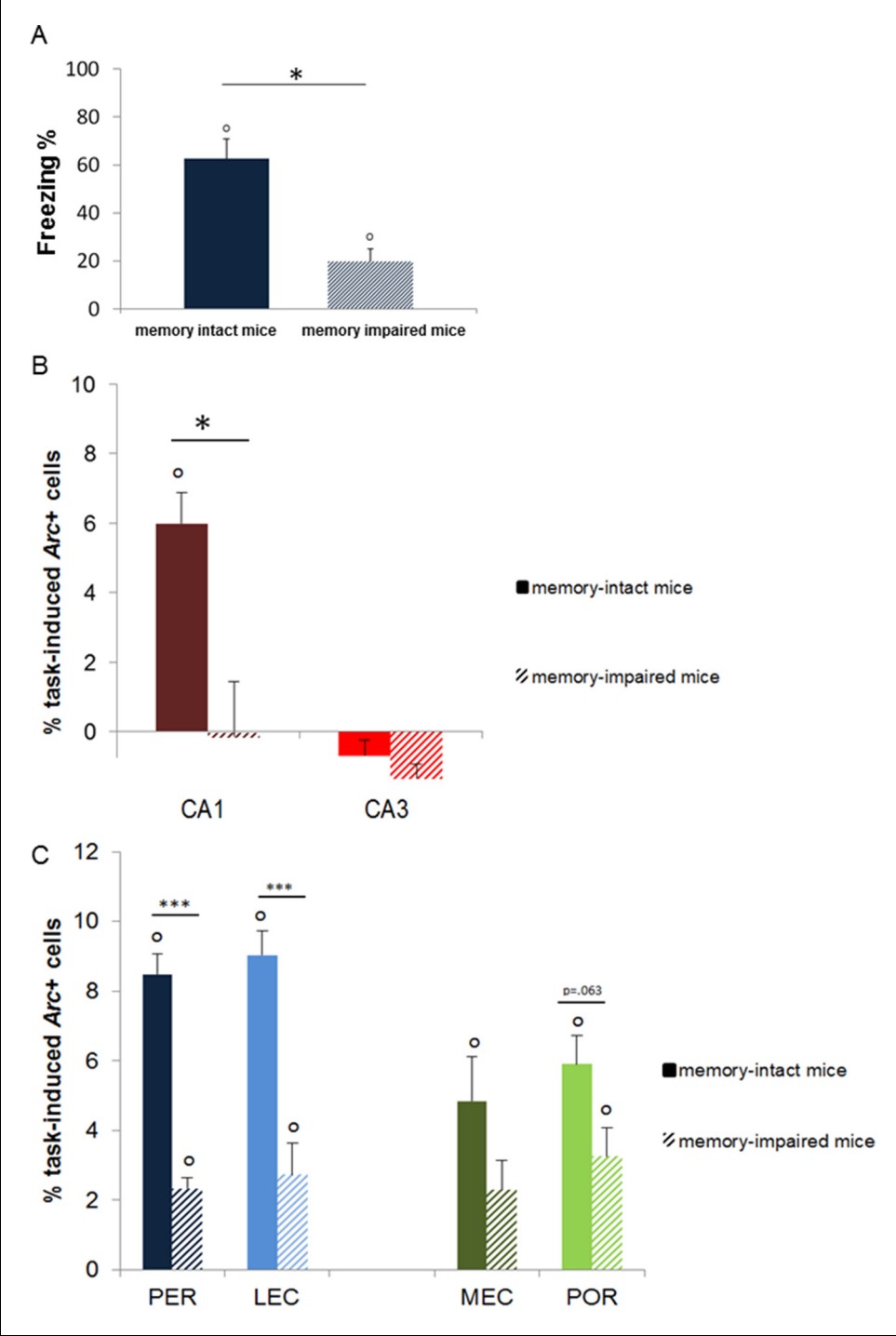

**Figure 4.** Memory performance of 'memory-intact' and 'memory- impaired' mice tested one year after conditioning and corresponding MTL patterns of activity. (**A**) Behavioral performance: 'memory-impaired' mice froze significantly less than 'memory-intact' mice at test, reflecting impaired memory retrieval in this group. (**B**) Activity patterns in CA1 and CA3: in contrast to mice that successfully retrieved the footshock-context association one year after conditioning, CA1 was not recruited in those that had impaired memory. In addition, activity levels in CA3 were comparable between the two groups, underlining the critical role of CA1 in the retrieval of very remote memories. (**C**) Activity patterns in parahippocampal areas: parahippocampal areas of the 'memory- impaired' mice were recruited to much lesser extent than those of 'memory-intact' animals. Since this reduced activation did not lead to successful memory retrieval, this result suggests that CA1 might play a role as important

*Figure 4 continued on next page*

*Figure 4 continued*

as the cortical areas in the retrieval of very remote memories in memory-intact mice. Error bars are mean ± SEM. 'o' indicate a significant comparison to 0 at p<0.05; asterisks a t-test at p<0.05 for '*' and at p<0.01 for '***'.

for this type of study, does not significantly differ between CA1 and CA3 (*Nakamura et al., 2013*). Thus, these findings suggest that the decline of expression seen in CA3 is specific to the task. These results show for the first time a clear functional segregation of CA3 and CA1 in the retrieval of memory over time, with CA3 contributing only to the retrieval of recent and early remote memories and CA1 being recruited independently of the age of the memory trace.

## The PER, the POR, the MEC, and the LEC are all maximally engaged for the retrieval of very remote memory traces

Second, we investigated the extent to which parahippocampal areas contribute to the retrieval of recent, remote and very remote memory traces. All parahippocampal areas participated qualitatively to a comparable extent to retrieving memory over time as they were all maximally engaged for the retrieval of the oldest traces. In addition, the PER and the LEC were more recruited than the POR and the MEC specifically for the oldest memory traces (PER vs LEC, PER vs POR, POR vs MEC and LEC vs MEC: delay effects: $Fs_{(4,30)}>18.89$, $ps<0.001$; within-area analysis; LEC, PER and POR: all $Fs_{(4,15)}>6.69$, $ps<0.003$; tukey posthocs: recent or early remote memories versus very remote memories: all $ps<0.006$; MEC: all $ps<0.047$ but for 1 month vs. 1 year: n.s; area effects: PER vs POR and LEC vs MEC: $Fs_{(1,30)}>10.91$, $ps<0.002$; comparisons for 6 months and one year: $ts_{(3)}>3.77$; $ps<0.034$ while PER vs LEC and POR vs MEC: n.s; interactions effects: PER vs POR and LEC vs MEC: $Fs_{(4,30)}>2.78$, $ps<0.045$ while PER vs LEC and POR vs MEC: n.s). In addition, all areas were activated across all delays (comparisons to 0: all $ts>3.52$, $ps<0.039$; *Figure 5A, B*) at the exception of the MEC and the POR for recent memories and the PER one week after conditioning (all $ts<2.22$, $ps>0.113$). However, this latter result might reflect a slightly larger SEMs for these time points rather than a true functional difference between brain areas since comparisons of activity levels between the LEC and the MEC and between the PER and the POR for the recent memory time points did not yield any significant differences ($ts_{(3)}<1.98$; $ps>0.142$). Of note, analysis with or without normalization of the task-induced *Arc* expression yields the same results. Interestingly, parahippocampal areas of animals showing impaired memory when tested one year after conditioning were also significantly activated (see *Figure 4C*), albeit less than those of animals whose memory was intact (comparisons to 0: memory intact: $ts_{(3)}>3.821$, $ps<0.032$; memory impaired: $ts_{(3)}>3.929$, $ps<0.029$, except MEC: $t_{(3)}=2.65$, $p=0.077$; intact vs. impaired: LEC and PER: $ts_{(6)}>5.58$, $ps<0.001$; POR: $t_{(6)}=2.27$, $p=0.063$; MEC: $t_{(6)}=1.67$, $p=0.146$). This result might suggest that, in the memory-impaired group, a partial memory representation had been consolidated and stored in the cortical areas, but that a weak activation of these areas alone does not suffice for successful retrieval. Thus, in summary, no qualitative differences were observed in the patterns of activation of the parahippocampal areas over time. However, the PER and LEC, which receive stronger connections from the amygdala than the MEC and the POR, were more activated than the MEC and the POR during the retrieval of the most remote memories (6 months- and one-year old traces), a time point which coincides with the epoch at which cortical areas are thought to play a strong role in the retrieval of memories. This result suggests a segregation of the parahippocampal areas in terms of memory processes rather than in terms of spatial or non-spatial information processing at this time point. In addition, these results further underline the critical role of CA1 in the retrieval of very remote memories as activation of the parahippocampal areas alone appeared not to be sufficient for a successful retrieval of the memory trace at this time point.

## Supplementary information: Analysis of the task-induced *Arc* expression with or without normalization (e.g. without subtracting the no-shock groups levels) yields the same results

Also, to focus on the fear response reflecting the context-footshock association (and not that contributed by other parameters), data presented in *Figures 3A*, *4B-C*, *5A-C* represent the task-induced *Arc* expression observed in the shock groups from which was subtracted that of the 'no-shock'

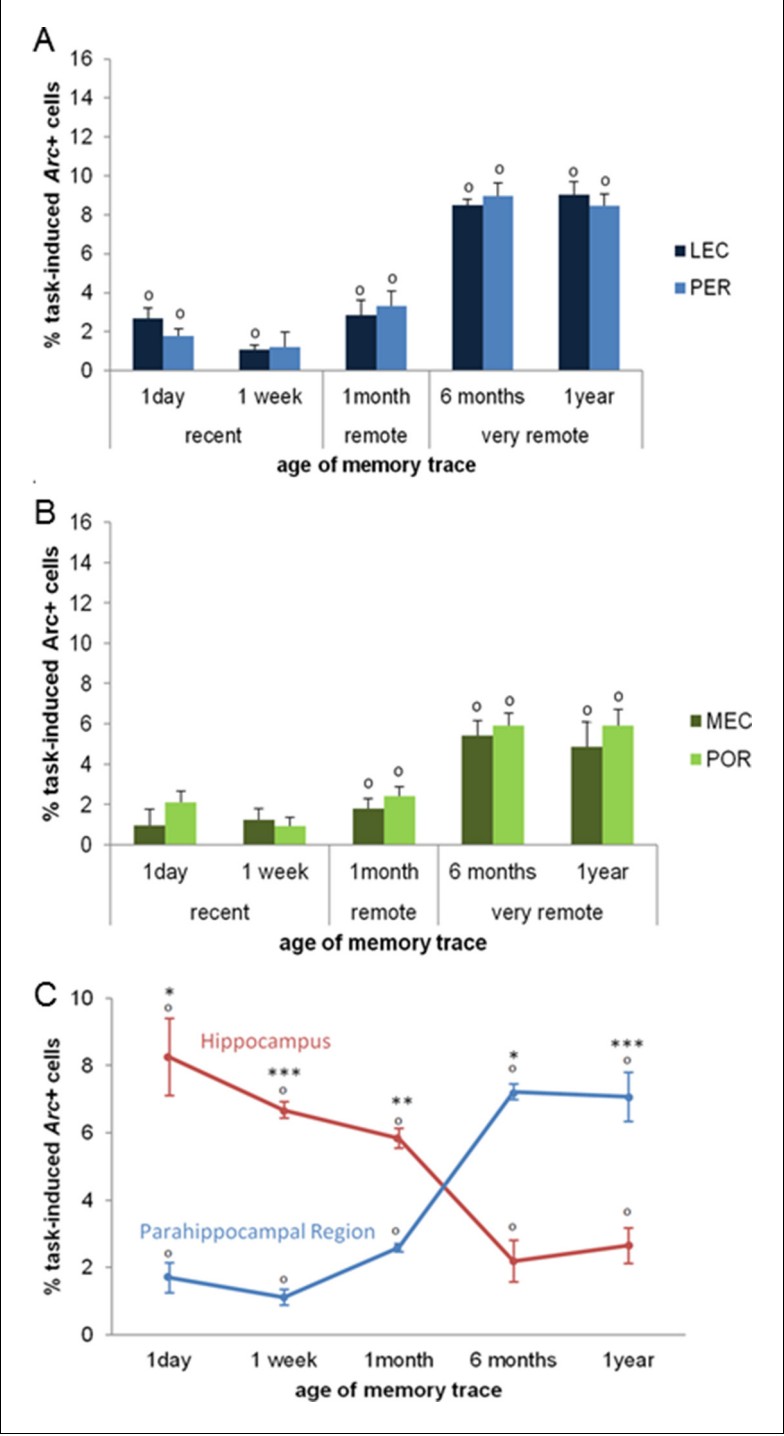

**Figure 5.** Activity patterns in the parahippocampal areas over time and apparent (and possibly misleading) over-time shift from the involvement of the hippocampus to the involvement of the parahippocampal region in memory retrieval. All parahippocampal areas were maximally engaged for the most remote memories and patterns of activity were comparable between the (A) LEC and PER and (B) the MEC and POR. Furthermore, in line with the existence of stronger projections from the amygdala to the PER and LEC than to the POR and MEC, and a more important role of the cortical areas for the most remote memories than for more recent ones, the levels of activity in the PER and the LEC were higher than those in the POR and MEC during the retrieval of very remote memories, providing further support to an emerging theory according to which the parahippocampal areas might be segregated in terms of memory types/processes rather than in terms of information content (spatial versus spatial information; *Eichenbaum et al., 2007*; of note, for the sake of clarity, significant area differences between graphs
*Figure 5 continued on next page*

*Figure 5 continued*

A and B are not shown). (**C**) Contribution of the hippocampal CAs and parahippocampal region to memory retrieval over time: when CA1 and CA3 activity levels are not dissociated, the overall contribution of the hippocampus to the retrieval of very remote memories is largely underestimated when compared to that of CA1 shown in *Figure 3A*. Even in this case though, the hippocampus is still significantly recruited at all times. In contrast, overall activity of the parahippocampal region is comparable to that of any of the parahippocampal areas, with a maximal activation during the retrieval of very remote memories (see *Figure 5A and B*). These results underline the need of dissociating CA1 and CA3 activity patterns to better understand the contribution of the hippocampus to the retrieval of memory over time. Importantly, when these contributions are not dissociated, the apparent over-time shift remains relative at most (and not absolute) since both areas are significantly activated at all delays. Error bars are mean ± SEM. 'o' indicate a significant comparison to 0 at p<0.05; asterisks a t-test at p<0.05 for '*' and at p<0.01 for '***'.

groups, which were subjected to the exact same experimental conditions but could not associate the context with an aversive experience (i.e. the footshock). Nevertheless, for the sake of transparency, the same analyses were also performed on the task-induced *Arc* expression found in the 'shock' and 'no-shock' groups. These analyses led the same outputs.

In the hippocampus, activity levels in the CA1 of the conditioned mice remained high at all delays while CA3 was no longer recruited for the retrieval of memories that were 6 months or older. Three-way ANOVA with 'shock', 'delay' and 'area' as factors and task-induced *Arc* expression as dependent variable showed that task-induced *Arc* expressions differed between CA1 and CA3 in function of the delay imposed and whether animals had been conditioned or not (shock effect: $F_{(2,90)}$ =519.74, p<0.001; delay effect: $F_{(4,90)}$=3.15, p=0.018, area effect: $F_{(1,90)}$=359.1, p<0.001; shock x delay effect: $F_{(8,90)}$=9.525, p<0.001; interaction shock x area effect: $F_{(2,90)}$, p<0.001; interaction area x delay effect: *n.s*; interaction shock x area x delay effect: $F_{(8,90)}$=3.63, p<0.001). Moreover, further separate two-way ANOVAs with 'shock' and 'delay' as factors on activity levels and t-tests between shock and no-shock groups in CA1 or CA3 that showed task-induced *Arc* expression in CA1 were higher in the conditioned groups than the no-shock groups for all delays, with the activation being the highest for the one day delay (CA1: shock effect: $F_{(1,30)}$=246.88, p<0.001; delay effect: $F_{(4,30)}$=1.79, n.s; interaction shock x delay effect: $F_{(4,30)}$=9.79, p<0.001; t-tests all ts>2.68, ps<0.036; one way ANOVA on CA1 activity levels in 'shock' groups: $F_{(4,15)}$=10.65, p<0.001, post-hoc Tukey: 1 day vs. all other delays ps<0.032; 6 months vs. 1 month p=0.046, all other *n.s*) while task-induced *Arc* expressions were higher in shock than in no-shock groups only for recent and early remote memories in CA3 (shock effect: $F_{(1,30)}$=83.38, p<0.001; delay effect: $F_{(4,30)}$=7.82, p<0.001; interaction shock x delay effect: $F_{(4,30)}$=13.35, p<0.001; t-tests shok vs. no shock: recent and early remote memories (t)s>6.14, ps<0.001; very remote memories (t)s<1.59, ps>0.161; one way ANOVA on CA3 activity levels in 'shock' groups: $F_{(4,15)}$=18.83, p<0.001; post-hoc Tukey: 1 year vs. recent and early remote memories: ps<0.001; 1 year vs. 6 months: n.s; 6 months vs. 1 day and 1 week ps<0.010; 6 months vs. 1 month p=0.076, all other *n.s*). In contrast, task-induced *Arc* expression in the 'no-shock' groups did not significantly differ accross delays in CA1 or CA3 (one way ANOVAs: CA1 $F_{(4,15)}$=1.92, *ns*; CA3 $F_{(4,15)}$=0.548, *n.s*).

In the parahippocampal region, the PER, LEC, MEC and POR are more recruited for the retrieval of 6 months-old and older memories. For the LEC and the PER, a three way-ANOVA with 'shock', 'delay' and 'area' as factors and task-induced *Arc* expression as dependent variable showed that the level of the recruitment of both areas is comparable and depends on the age of the memory trace and wether mice received a footshock or not (shock effect: $F_{(2,90)}$=465.9, p<0.001: delay effect: $F_{(4,90)}$=37.75, p<0.001, area effect: *n.s*; interaction effects: shock x area, area x delay and shock x area x delay: *n.s*; interaction shock x delay effect: $F_{(8,90)}$=34.46, p<0.001). These results were confirmed by two-way ANOVAs performed separately on LEC and PER activity levels (LEC: shock effect: $F_{(2,45)}$=235.86, p<0.001; delay effect: $F_{(4,45)}$=17.18, p<0.001; interaction shock x delay effect: $F_{(8,15)}$=16.98, p<0.001; PER: shock effect: $F_{(2,45)}$=230.04, p<0.001; delay effect: $F_{(4,45)}$=21.39, p<0.001; interaction shock x delay effect: $F_{(8,15)}$=17.82, p<0.001). Finally, further within group one-way ANOVAs in the LEC or the PER revealed that the delay at which these areas were the most recruited in the conditionined mice were 6 months and one year, while such an effect was not observed in the no-shock groups (LEC: shock groups: $F_{(4,15)}$=30.14, p<0.001, post-hoc Tukey:

6 months and 1 year vs. all other delays: ps<0.001, all other ps: *n.s.*; no-shock groups: F(4,15) =0.875, *n.s.*; PER: shock groups: F(4,15)=45.98, p<0.001, post-hoc Tukey: 6 months and 1 year vs. all other delays ps<0.001; 1 month vs. all other delays ps<0.030, all other ps: n.s.; no-shock groups: F(4,15)=1.15, n.s.)

The patterns of activity in the MEC and the POR were comparable and linked to the age of the memory trace and wether mice were conditionined or not (three way-ANOVA: shock effect: F(2,90) =174.51, p<0.001: delay effect: F(4,90)=9.49, p<0.001, area effect: F(1,90)=2.05, n.s; interaction effects: shock x area, area x delay and shock x area x delay: *n.s*, interaction shock x delay effect: F (8,90)=10.94, p<0.001). Results that were confirmed by two-way ANOVAs performed separately on MEC and POR activity levels (MEC: shock effect: F(2,45)=66.35, p<0.001; delay effect: F(4,45)=3.45, p<0.015; interaction shock x delay effect: F(8,15)=4.31, p<0.001; POR: shock effect: F(2,45)=117.22, p<0.001; delay effect: F(4,45)=6.98, p<0.001; interaction shock x delay effect: F(8,15)=7.34, p<0.001). Lastly, further within group one-way ANOVAs performed separately on the MEC and the POR activity levels revealed that the delays at which these areas were the most recruited in conditioned mice were the 6 months and one year delays, while such an effect was not observed in animals that did not received a shock at conditioning (MEC: shock groups: F(4,15)=6.32, p<0.005, post-hoc Tukey: 6 months vs. all other delays: ps<0.028; 1 year vs. 1 day p=0.049; 1 year vs. 1 week and 1 month p=0.088 and 0.080, respectively; no-shock groups: F(4,15)=0.167, *ns*; POR: shock groups: F (4,15)=18.41, p<0.001, post-hoc Tukey: 6 months and 1 year vs. all other delays ps<0.007; all other ps: ns; no-shock groups: F(4,15)=0.304, *ns*)

### A relative and potentially misleading over-time shift of a stronger engagement of the hippocampus to a stronger involvement of the parahippocampal region in the retrieval of memory

To compare more readily our results to the vast majority of studies in humans which do not dissociate the contribution of CA1 from that of CA3 to memory retrieval, nor that of the different parahippocampal areas, we also analyzed the overall activity of the hippocampal CAs and that of the parahippocampal region by averaging CA1 and CA3 activity levels on the one hand and those of the parahippocampal areas on the other hand (*Figure 5C*). Activity in the hippocampus was higher than that of the parahippocampal region for the retrieval of recent and early remote memories while an inversed relationship emerged for the retrieval of very remote memories (area effect: F(1,30) =9.984; p=0.004; delay effect: F(4,30)=1.539; p=0.216; delay x area interaction effect: F(4,30) =61.629, p<0.001; post-hoc analysis: recent and early remote memory: ts>4.21, ps<0.024; very remote memories: ts>6.07 ps<0.009). Importantly, even under this condition, e.g. when the contribution of CA1 was not dissociated from that of CA3, the hippocampus was significantly recruited independently of the age of the memory trace (comparisons to 0: all ts>3.54 ps<0.038) in line with the multiple trace theory. Furthermore, in this case, the hippocampus appeared to contribute mainly to retrieving recent and early remote memories (F(4,19)=16.21, p<0.001; Tukey posthocs: recent or early remote memories vs very remote memories: all ps<0.024). Conversely, the parahippocampal region was maximally activated for retrieving very remote memories (F(4,19)=52.51, p<0.001; recent or early remote memories vs very remote memories: all ps<0.001) but was also recruited for recent and early remote memories (comparisons to 0: ts(3)>3.79; ps<0.032). In conclusion, this analysis underlines the fact that, if CA1 and CA3's contribution are not dissociated, the hippocampal contribution to the retrieval of very remote memories might be largely underestimated. Nevertheless, even in this case, the over-time shift from an engagement of the hippocampus to that of an engagement of the parahippocampal region in retrieving memory is only relative and not absolute, given that the hippocampus is still significantly activated even for the retrieval of very remote memories.

## Discussion

Investigating memory retrieval over such an extended period of time enabled us to provide the first evidence of a time-limited contribution of CA3 to the retrieval of memory while, in striking contrast, CA1 remained involved regardless of the age of the memory trace. In addition, parahippocampal areas were substantially more involved in retrieving very remote memory traces than more recent ones and their activity patterns appeared to reflect memory demands rather than information content.

To our knowledge, no study has investigated the role of CA3 in retrieving remote memory traces in humans and only one studied early remote memory in animals (e.g. up to one month-old traces which roughly corresponds to a 4 year-old memory trace in humans based on life expectancy; *Gusev et al., 2005*). Thus, the contribution of CA3 to the retrieval of memory traces as old as what is considered to be a remote/very remote memory trace in humans (in the order of decades) is not known. For this reason, we studied 6 months-and one year- old memories traces in mice, which roughly correspond to 20 and 40 year-old memory traces in humans. Doing so enabled us to report for the first time the lack of engagement of CA3 in the retrieval of very remote memories. This time-limited role of CA3 was supported by a lack of recruitment during the retrieval of the most remote memory traces (6 months- and one year-old) while CA3 was strongly activated for recent and early remote memories (up to 1 month-old traces). This result was confirmed by a significant correlation between CA3 activity levels and memory performance during the retrieval of the recent and early remote memories, which did not hold for very remote traces. The approach we have adopted does not allow for the characterization of the specific process by which CA3 fails to aid in retrieving memory. However, since computational studies and studies that have limited their investigations to recent and early remote memory traces have speculated that CA3's contribution to memory retrieval would increase as memory ages because of an increasing demand on the completion of representations using partial or degraded features of these stored representations (the 'pattern completion theory': *Rolls et al., 1997*), we speculate that the lack of engagement of CA3 in retrieving very remote memories stems from a failure to 'pattern complete' memory representations. Indeed, by this time, these very old memory representations might have degraded to such an extent that it is impossible to use any specific features as an efficient retrieval cue. This hypothesis would go along with reports demonstrating that details of memory representations are lost as memory ages (*Wiltgen and Silva, 2007*; *Wiltgen et al., 2010*) and will require further investigations to be tested.

In striking contrast to the limited involvement of CA3, we found an ongoing involvement of CA1 in memory retrieval. This result is in agreement with the only two studies specifically investigating the role of CA1 in remote memory retrieval. Indeed, transient inhibition of cell firing in CA1 in mice using optogenetics drastically impaired fear memory retrieval, one or twenty eight days after conditioning (*Goshen et al., 2011*). Likewise, a recent fMRI study in humans showed impairments in autobiographic and episodic memory for traces even older than 30 years-old following focal CA1 lesions (*Bartsch et al., 2011*). Our result of a selective involvement of CA1 in the retrieval of very remote memories at this time point is supported by the lack of a recruitment of this area in the memory-impaired group tested one year after conditioning as opposed to a strong activation in the memory intact group and the fact that CA3 was not recruited in either group (*Figure 4B*). Importantly, using a high resolution molecular imaging technique, we bring the first evidence that the persistent engagement of the hippocampus in memory retrieval does not involve CA3. This result, in turn, suggest that hippocampal patterns of activation observed in previous human fMRI studies investigating the retrieval of very remote memory, might essentially be driven by CA1, but could not be specifically identified as such because of the limited spatial resolution of standard fMRI techniques (*Ryan et al., 2001*; *Rekkas and Constable, 2005*; *Bonnici et al., 2012*). Moreover, the fact that only CA1 is activated for these late time points suggests a preferential involvement of the temporoammonic pathway over that of the trisynaptic loop for the retrieval of very remote memories. Robust evidence for this hypothesis is not yet available for very remote memories, but lesions of the temporoammonic pathway in rats specifically impaired early remote memory (28 days-old) in rats tested in a Morris watermaze task, while recent memory (one day-old) was spared. This result suggests that direct entorhinal cortex inputs to CA1 contribute to the retrieval of early remote memory, possibly by further consolidating memories that were not consolidated after 28 days (*Remondes and Schuman, 2004*). In the present study, because we investigated a much larger time-window and found that CA1 was still activated in addition to the parahippocampal cortical areas, but not CA3, we speculate that the temporoammonic pathway is not only important for the consolidation of memories but also for the retrieval of already consolidated memories.

Furthermore, parallel to the disengagement of CA3 in the retrieval of very remote memories, we found a maximal involvement of the parahippocampal areas. The hypothesis of a functional segregation of the parahippocampal region between a 'PER-LEC stream' and a 'POR-MEC stream' is essentially based on studies investigating the role of the PER during the retrieval of recent memory, while much fewer have addressed the specific role of the POR, the LEC and the MEC within this frame and

during the retrieval of early remote memories. In addition, no study has yet investigated their contribution for very remote memories, despite the belief that the contribution of cortical areas to memory retrieval would be the largest for the most remote memories. Our results for recent and early remote time-points match published data according to which, for example, the PER and the POR are recruited to a similar extent for contextual fear conditioning and the LEC and MEC contribute to a similar extent to memory retrieval (*Bucci et al., 2002*; *Burwell et al., 2004*; *Izquierdo et al., 1997*; *Goshen et al., 2011*). In addition, we show for the first time that the contribution of the PER, the POR, the LEC and the MEC is indeed the largest for the retrieval of very remote memories. Patterns of activation were comparable between the LEC and the PER and between the MEC and the POR. Interestingly though, LEC and PER were more activated than the MEC and the POR during the retrieval of very remote memory traces. This result appears to rather reflect the type of memory process the areas were engaged into (fear conditioning) rather than the content of information processed (contextual information), supporting a view that has recently emerged from the literature (*Eichenbaum et al., 2007*; *Beer et al., 2013*). Indeed, the amygdala which plays a crucial role in fear conditioning, sends stronger projections to the PER/LEC than to the MEC/POR (*Pitkänen et al., 2000*) and higher activity levels in the POR and MEC than in the PER and LEC would have been expected if information content was primarily reflected in the pattern of activation observed. Altogether, our data show a preferential role of the parahippocampal region for the retrieval of very remote memories as opposed to that of early remote and recent memories, matching the pattern commonly reported for the prefrontal cortex (*Frankland and Bontempi, 2005*).

In summary, our results provide the first evidence of a disengagement of CA3 for the retrieval of very remote memories and identify CA1 as the main player at this time point. These results are consistent with the view that at least the hippocampal subfield CA1 is involved in retrieving memory independently of the age of memory trace, possibly by providing support to cortical areas that include the parahippocampal areas. Of note, averaging activity levels of the hippocampal areas on the one hand and those of the parahippocampal areas on the other hand revealed that both the hippocampus and the parahippocampal region were activated independently of the age of the memory trace (*Figure 5C*). Hence, the gradual "transfer" of the memory trace from the hippocampus to the cortex is unlikely to be an all-or-nothing phenomenon, but seems to go along with a shift in the relative contribution of the two regions as the memory trace ages, and possibly a shift in the contribution of the networks they belong to. More specifically, our results suggest that the trisynaptic loop could play a preponderant role in retrieving early memories as reflected by an involvement of CA3, CA1 and the parahippocampal areas at this stage, while the retrieval of very remote memories would require an heavier contribution of the temporoamonic pathway as indicated by the lack of recruitment of CA3 and a strong activation of CA1 and the parahippocampal areas.

## Materials and methods

### Animals
8–12 weeks old male C57BL/6 mice (n=72 total) bred at the Ruhr-Universität Bochum were used. The age of the animals at the term of the experiment varied between 8 and 70 weeks. Animals were housed in groups of 2–3 animals, kept under reversed 12-hr light/dark cycle (8:00 a.m. light off; 8.00 p.m. light on) and tested during their active phase. Access to food and water was ad libitum and all procedures were approved by the Ruhr-Universität Bochum Institutional Animal Use Committee and the LANUV (84-02.04.2013.A419)

### Behavioral paradigm
#### Apparatus
The fear-conditioning setup consisted of a 26×26×38 cm arena with a stainless steel grid floor and plexiglass walls. A dim light and a speaker that delivered white noise (55 dB) were attached to the walls and the conditioning chamber was placed into a soundproof box with a lid. Two infrared sensor rings fitted the arena and detected vertical and horizontal activity using the TruScan photobeam activity system (Coulbourn Instruments, Allentown, PA, USA)

## Procedure

Mice were randomly divided into three groups (shock, no-shock and home-cage). For the four first delays (1 day, 1 week, 1 month and 6 months) four mice were tested per group (e.g. n=12 per delay; n=48 total). Since preliminary studies showed that approximately half of the mice tested one year after conditioning had 'forgotten' the context-footshock association, a confounding factor for the imaging data, twice as many mice were conditioned for the one year 'shock' group (n=8 for the 'shock' group; n=16 total for this delay). To test whether the fear conditioned response observed in shocked animals was specific to the context in which they were conditioned, two additional groups of mice (n=4 each; n=8 total) were conditioned in the usual conditioning box and later tested in a completely novel context differing in olfactory, tactile and visual cues (lightening, texture of the floor and background noise) instead of the original conditioning arena. Animal were tested as triplets formed of one 'shock', one 'no-shock' and one 'home-cage' mice.

The fear-conditioning protocol was adapted from *Sauvage et al. (2000)*. On conditioning day, mice belonging to the 'shock' groups were placed in the conditioning chamber that they could freely explore for 2.5 min (baseline). Subsequently, a 1 mA footshock was delivered for 2 s and mice remained for another 2.5 min in the arena. 'No-shock' control animals underwent the exact same procedure, except that they did not receive a shock. 'Home-cage' controls stayed in their cages at all times and were used to evaluate baseline *Arc* expression. The arena was thoroughly cleaned with distilled water and 10% ethanol after each animal. After a delay of either 1 day, 1 week, 1 month, 6 months or 1 year, the memory for the association of the context with the foot-shock was tested by re-exposing the mice to the same conditioning chamber for 6 min while no shock was delivered (test day).

## Behavioral analysis

The cumulative time during which an animal did not change its position (resting time) during the sampling time of 250 ms was measured both during conditioning and test. Conditioned suppression of activity, e.g. relative resting time or freezing index, was taken as indicator of fear conditioning (*Sauvage et al., 2000*; *Rammes et al., 2000*). Because the baseline resting time of the 'shock' one-year group (one of the ten groups studied) was slightly lower (or trending to be) than that of the other shock groups ($F(4,15)=3.37$, $p=0.037$; post-hoc Tukey: 1 year vs. 1 day $p=0.041$; vs. 1 month or 6 months ($p=0.079$ and $0.071$, respectively) and because a similar pattern was found at test, focusing solely on the% resting time at test could have led to a misinterpretation of the test data since the lower freezing index at test is carried by a lower baseline at conditioning. Hence, freezing levels discussed in the manuscript have been normalized by the conditioning baseline for each group: the freezing index was calculated by normalizing the resting time during the first 2.5 min of test by the baseline resting time during the first 2.5 min of conditioning: freezing index= ([test resting time- baseline resting time]/baseline resting time)*100. Also, to focus as much as possible on the conditioning suppression activity related to the context-footshock association (and not that contributed by other parameters), differences in freezing indices between the 'shock' and 'no-shock' groups are represented in *Figure 1B*. These were calculated by subtracting the freezing indices of the 'no-shock' groups from that of their age-matched 'shock' controls. Importantly, statistical analysis of the data with or without baseline or 'no-shock' normalizations (e.g.% resting time at test) yield similar results. Finally, as predicted by pilot experiments and suggested by studies in humans (*Lindenberger, 2014*), about half of the one-year 'shock' group showed poor memory performance: freezing levels of half of the one-year 'shock' group (n=4) were quite low ($24.76 \pm 2.2\%$ in average as opposed to $62.85 \pm 5.4\%$ for the other half of the group). This group is referred to as 'memory-impaired' group in the manuscript and their performance underwent a separate analysis (see *Figure 4A–C*).

## Imaging activity in the medial temporal lobe

Detection of the immediate-early gene *Arc* using fluorescent in situ hybridization (FISH) techniques. Animals were sacrificed by decapitation immediately after the test session. Brains were removed, flash frozen with iso-pentane and stored at -80°C. Frozen brains were sectioned into 8 µm thick sections with a cryostat (Leica CM 3050 S) and mounted on Polylysine slides (Thermo Scientific), which were stored at -80°C. T. Kitsukawa (Osaka University, Japan) provided the plasmid for generating *Arc* antisense and sense RNA probes. The plasmid contained a ~1.2 kbp *Arc* transcript including

intron regions. A digoxigenin-labeled UTP kit (Roche Diagnostics) was used to synthesize *Arc* pre mRNA probes. A FISH protocol adapted from *Guzowski et al. (1999)* was used. Briefly, brain sections were fixed with 4% buffered paraformaldehyde and rinsed with 0.1 M PBS. After treatment with an acetic anhydride/triethanolamine/hydrochloric acid mix they were rinsed again and soaked with a prehybridization buffer. The slides were then hybridized with a digoxigenin-labeled antisense *Arc* intron-enriched riboprobe overnight at +65°C. After rinses with SSC buffer solutions, the slides were treated with an anti-digoxigenin-horseradish-peroxidase (HRP) conjugate (Roche Molecular Biochemicals). The signal was amplified using a cyanine-5 substrate kit (CY5, TSA-Plus system, Perkin Elmer). Nuclei were counterstained with 4′, 6′-diamidino-2phenylindole (DAPI; Vector Laboratories). As a control for probe specificity, extra slides were hybridized without probes or with *Arc* antisense probes. There was no *Arc* labeling detectable in those slides.

### Image acquisition and evaluation of *Arc* expression

To detect *Arc* RNA expression in all areas of interest, three slides per animal were processed and analyzed. Each slide contained 6 nonconsecutive brain sections (distant ~50 μm). One slide contained CA1 and CA3 (~-2.5 mm AP), another slide LEC and PER (~-3 mm AP) and another MEC and POR (~-4.5 mm AP)(*Paxinos and Franklin, 2004*). Six images from these nonadjacent sections were acquired for each area of interest and covered ~900 μm. A Keyence Fluorescence Microscope (BZ-9000E; Japan) was used to capture images. The images had 1360x1024 pixels (362 x 272 μm$^2$) and were taken with a 40x objective (z-stacks of 0.7 μm thick pictures). Images were taken with constant exposure time, light intensity and contrasts optimized for the appearance of intranuclear labeling as described in *Vazdarjanova and Guzowski (2004)*. To account for stereological considerations, counting was performed on 8 μm thick nonadjacent sections that contained one layer of cells (*West, 1999*). Cells were counted in a double-blind manner with the experimenter being unaware of the hypothesis tested. Cells were counted as *Arc* positive when one or two characteristic red dots were visible in the DAPI labeled nucleus. Cells without these dots were counted as negative. The percentage of *Arc* positive cells was calculated as: number of *Arc* positive cells/(number of *Arc* positive cells + number of *Arc* negative cells) x 100. "Task-induced *Arc* expression" was calculated for each MTL area by subtracting *Arc* expression found in no-shock controls from the *Arc* expression found in animals that had received a shock during conditioning. Because all experimental conditions were kept identical and only the delay between conditioning and test differed, comparisons of task-induced *Arc* expression are thought to reflect differences in fear memory retrieval that dependent upon the age of the memory trace.

## Statistical analysis

### Behavioral performance

A one way ANOVA was used to compare freezing levels across delays and one-sample t-tests for comparisons to zero. Independent sample t-tests were used to compare freezing to the conditioning context versus freezing to the new context between the 'shock' and 'no-shock' groups 1 day or 1 year after conditioning and behavioral performance between the memory-intact and memory-impaired groups.

## Imaging

To reduce type 1 error and keep type 2 errors low, only a priori hypotheses were tested, e.g. statistical analysis were restricted to the comparisons of task-induced brain activity in and between CA1 and CA3, PER and LEC or POR, MEC and POR or LEC, the hippocampal CAs and the parahippocampal region. For these comparisons two-way ANOVAs with 'area' and 'delay' as factors and 'task-induced *Arc* expression' as the dependent variable were used. In addition, one way ANOVAs followed by tukey post-hocs were performed for within-areas analysis and independent paired t-tests for between-areas comparisons.

## Acknowledgements

Supported by the Mercator Stiftung, the SonderForschungBereich 874 and the International Graduate School for Neuroscience (IGSN) of the RUB.

## Additional information

### Funding

| Funder | Author |
| --- | --- |
| Stiftung Mercator | Vanessa Lux<br>Magdalena M Sauvage |
| International Graduate School for Neuroscience (IGSN) of the RUB | Vanessa Lux |
| SonderForschungBereich 874 | Erika Atucha<br>Magdalena M Sauvage |

The funders had no role in study design, data collection and interpretation, or the decision to submit the work for publication.

### Author contributions

VL, Conception and design, Acquisition of data, Analysis and interpretation of data, Drafting or revising the article; EA, Analysis and interpretation of data, Drafting or revising the article; TK, Analysis and interpretation of data, Drafting or revising the article, Contributed unpublished essential data or reagents; MMS, Conception and design, Analysis and interpretation of data, Drafting or revising the article

### Author ORCIDs

Magdalena M Sauvage, http://orcid.org/0000-0002-7586-6410

### Ethics

Animal experimentation: all procedures were approved by the Ruhr-Universität Bochum Institutional Animal Use Committee and the LANUV (84-02.04.2013.A419).

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
