## [Decision Letter]

Thank you for submitting your work entitled "Imaging a memory trace over half a life-time in the medial temporal lobe reveals a time-limited role of CA3 in retrieval" for consideration by *eLife*. Your article has been favorably evaluated by Timothy Behrens (Senior editor) and three reviewers, one of whom is a member of our Board of Reviewing Editors. One of the three reviewers, Paul Frankland, has agreed to share his identity.

The reviewers have discussed the reviews with one another and the Reviewing editor has drafted this decision to help you prepare a revised submission.

Summary:

It is widely accepted that memories for events are encoded in hippocampal-cortical networks, but over time these memories become less dependent onthe hippocampus. There do remain several complexities and controversies aboutthis idea. This paper reports a study that imaged brain activity related tocontext fear conditioning in mice after retention intervals ranging from

1 day to 1 year using the immediate-early gene *Arc*. The study reportshow different regions of the hippocampus and parahippocampal areas changetheir involvement in memory over this very wide range of times.

Essential revisions:

1) The reviewers agreed that the study is a valuable look at the role ofhippocampus in very long-term memory recall. The attention to hippocampalsubdivisions is important.

2) All the reviewers expressed concerns about the statistical analysis.

These included the sample size, the methodology of the analysis (controls and double-blind), and the need to tighten up the analysis of the CA1-CA3 dissociation. The authors should address these points in the detailed comments below.

3) The analysis of correlation between *Arc* and memory performance should also be improved.

4) The authors should better place their study and its findings in contextof existing work.

*Reviewer #1:*

This study examines cellular and behavioral signatures of memoryfor up to a year (roughly half a lifetime for a mouse). The authors report thatthe CA3 no longer retains cellular signatures of memory at long times,but the CA1 and various parahippocampal regions do. This is an interestingfinding, with implications for the differential role of CA1 and CA3 invery long-term memory storage in the mouse.

Major points:

1) Is there a positive control for stimulus-evoked *Arc* expression in theseregions? Possibly there is an age-related decline in such expression in CA3but no other regions. The authors should provide citations that addressthis point, otherwise the concern would be that the decline ofexpression is not specific to the task.

2) Was the identification of *Arc* labeling automated? From the text it appearsthat this was not so. Then, were the slices counted in a double-blindmanner? This does not seem to have been the case, but the authors shouldclarify. If it was not double-blind then this is something of a concern.

*Reviewer #2:*

Memories for events etc. are initially thought to be encoded in hippocampal-cortical networks but, over time, become less dependent on the hippocampus for their expression. While this idea has reached textbook status, it is not without controversy/complexities. For example, in rodent studies, the hippocampal dependency at remote time points may depend upon memory quality (e.g., Wiltgen studies). Moreover, in some cases there is evidence that the hippocampus never really becomes 'unimportant' (e.g., Sutherland studies). Using a molecular imaging approach to examine patterns of activation following recall of a contextual fear memory (imaging for *Arc* mRNA as an indicator of recently activated neurons), the current paper addresses this now classic question, but with 2 important twists. First, the authors pay attention to hippocampal subdivisions (and do not consider the hippocampus as a whole) and, second, they use very remote time-points to probe memory organization (i.e., 6 months to a year instead of the standard one month 'remote' time-point). Both these novel aspects of the design, I believe, have the potential to offer new insights.

The core findings: 1) CA1 remains 'engaged' during memory recall regardless of memory age; 2) CA3 becomes disengaged at only the very remote time-points (6 months, 1 year); 3) Parahippocampal regions become increasingly engaged with memory age. The important take home is the CA1/CA3 dissociation at the very remote time points.

Overall strengths: The CA1/CA3 dissociation is important. In previous rodent imaging studies shorter time frames have been used and the hippocampus has typically not been subdivided (but see below). This dissociation may explain some of the 'noise' in the literature, in much the same way as Wiltgen's distinction between hippocampal engagement during recall of precise vs imprecise context memories. Finally, the inclusion of parallel analyses of transformed/non-transformed data sets is a strength.

Overall weaknesses: The most interesting aspect of the paper is the CA1/CA3 dissociation at very remote time-points. However, the statistical support for this isn't as clear cut as one might hope. Comparing CA1 and CA3 activation directly, they differ at 1 month and 1 year, but not at 6 months (Figure 3). A related concern is the necessity to drop animals from analysis at the 1-year time point likely muddies the interpretation of this time point. A second weakness, when placing this paper in the broader context of recent studies examining organization of context fear memories, is the lack of causal intervention (e.g., optogenetic silencing of tagged 'engram' cells in Tonegawa, Mayford, Inokuchi and Wiltgen papers).

Specific comments:

1) The sample sizes are low (n=4 per group) for these types of analyses, and may contribute to the lack of statistical clarity with respect to the CA1/CA3 dissociation.

2) It is not clear to me why freezing levels would be predicted to decline in the 1-year group – is there a citation the authors could include? The only paper that I am aware of-Gale et al. (2004) did not observe a drop off at super long delays.

3) The authors did not see a time-dependent increase in generalization. This is quite unexpected given many previous studies (e.g., Wiltgen), and is worthy of more comment and discussion. In fact, this idea of time-dependent generalization is brought in the Discussion (second paragraph) and the authors use it to explain the absence of CA3 activation (because representation is degraded sufficiently to make pattern completion not possible). But they don't see any evidence for this in their behavior.

4) Correlation analysis. I found these quite unconvincing. To support the argument that memory performance and *Arc* levels are related the authors need to find a correlation within the shocked group. The current analysis only really shows that shocked animals have more *Arc*.

*Reviewer #3:*

The paper reports on a study that imaged brain activity related to context fear conditioning in mice after retention intervals ranging from 1 day to 1 year using the immediate-early gene *Arc*. The involvement of CA1 and CA3 decreased across delays, whereas the involvement of the parahippocampal regions increased across delays. The results were taken as indicating that in contrast to CA1, CA3 is no longer recruited in very remote retrieval, and that parahippocampal areas showed increasing involvement as delay increased.

Overall, I think the results are novel and that the methods are sound. So I think this will make a nice contribution to the literature. But I did have a few suggestions.

1) The dissociation between the hippocampal regions and the parahippocampal regions seems well supported by the results. However, the apparent dissociation between CA1 and CA3 is not nearly as compelling. It seems to me that both regions are involved in retrieval and they both decrease across delays, it’s just that CA3 starts off lower and so it approaches nonsignificance earlier. In order to support a claim of a dissociation here it would be necessary to show a significant interaction between regions (CA1 va CA3) and forgetting rate. As it is, it’s just that at the longer delays CA3 activity is not significantly above threshold whereas CA1 activity is. Given CA1 was always showing a larger memory effect it is perhaps not too surprising. I think the other dissociation is very interesting, so the results will be of considerable interest to many researchers, however I think that the discussion of the CA1 – CA3 dissociation needs to be toned down (unless the interaction is significant).

2) I found the Introduction to be a bit confusing. For example, I did not feel like I had a good idea of what the existing literature has shown us with respect to the involvement of these MTL regions in studies of fear conditioning across these types of retention intervals. Moreover, it was not entirely clear why the specific delays were examined in the current study.

---

## [Author Response]

Reviewer #1:

*1) Is there a positive control for stimulus-evoked Arc expression in theseregions? Possibly there is an age-related decline in such expression in CA3but no other regions. The authors should provide citations that addressthis point, otherwise the concern would be that the decline ofexpression is not specific to the task.*

We have shown in a previous study that *Arc* expression elicited by a maximal electroconvulsive shock (a standard positive control for such type of studies) did not significantly differ between CA1 and CA3 (Nakamura et al., J. of Neuroscience, 2013). In addition, in the present study, we established the absence of age-related decline in baseline *Arc* expression specific to CA3 (see Figure 6). Thus, these findings suggest that the decline of expression seen in CA3 is specific to the task.

Author response image 1.we have now integrated these points in the results section of revised manuscript b adding following sentence.**DOI:**
http://dx.doi.org/10.7554/eLife.11862.008

“Of note, these results are unlikely due to an age- related decline specific to CA3 since baseline *Arc* expression was comparable between and within CA1 and CA3 over time as shown by the absence of significant ‘area’ x ‘delay’ interaction, main area effects and posthocs analysis over time failing to reach significance (area effect: (F(1,30)= 2.30; p=.14), delay effect: F(4,30)=.77;p=.55, area by delay interaction effect: (F(4,30)=40;p=.81; posthocs: delay effects: CA1: F(4,20)=.63, p=.65; CA3: F(4,20)=.52, p=.72) and because *Arc* expression elicited by a maximal electroconvulsive shock, a standard positive control for this type of study, does not significantly differ between CA1 and CA3 (Nakamura et al., 2013). Thus, these findings suggest that the decline of expression seen in CA3 is specific to the task”.

*2) Was the identification of Arc labeling automated? From the text it appearsthat this was not so. Then, were the slices counted in a double-blindmanner? This does not seem to have been the case, but the authors shouldclarify. If it was not double-blind then this is something of a concern.* Cells were counted in a double-blind manner with the experimenter being unaware of the hypothesis tested. This is now stated in the Materials and methods as follows: “Cells were counted in a double-blind manner with the experimenter being unaware of the hypothesis tested”.

Reviewer #2:

*[…] Overall weaknesses: The most interesting aspect of the paper is the CA1/CA3 dissociation at very remote time-points. However, the statistical support for this isn't as clear cut as one might hope. Comparing CA1 and CA3 activation directly, they differ at 1 month and 1 year, but not at 6 months (Figure 3).*

Reviewer 2 does not ask us to address this comment at this level but we feel it is important to underline here that CA1 and CA3 activation also differs at 1 day and that even though a direct comparison (t-test) at 6 months fails to reach significance, there is a clear visual trend for such a difference at this delay. Moreover, as a further indirect support for a difference between the two areas at 6 months, only activity in CA1 differs from 0 (i.e. only CA1 is engaged during the retrieval of memory 6 months after conditioning) and not that of CA3. To boot, a more stringent statistical analysis of the data using a 2-way ANOVA with ‘areas’ and ‘delays’ as factors revealed a significant area by delay interaction effect showing quite robustly that the contribution of CA1 and CA3 in retrieving memory over time differs. We have answered this comment in more detail below.

*A related concern is the necessity to drop animals from analysis at the 1-year time point likely muddies the interpretation of this time point.*

Here again the Reviewer 2 does not ask us to address this point. However, since there could have been a misunderstanding at this level, we would like to point out that no animals were ‘dropped’ from the 1 year time point but that (as mentioned in the original manuscript Materials and methods; procedure) activity patterns of animals that remembered the association at this time point (memory-intact mice) and those of mice that did not (memory-impaired mice) were analyzed separately to prevent for differences in memory strength between delays to confound the imaging data. This way, on the one hand, activity levels in groups with comparable accuracy were compared over time (Figure 3) and on the other hand, activity levels in mice with impaired or intact memory after one year were compared (Figure 4). Importantly, the latter comparison gave further support to our main finding (Figure 3): that CA1 is crucial for the retrieval of very remote memory but not CA3, since activity differed only in CA1 between memory-impaired and memory- intact mice. In other words, the interpretation of the 1-year time point would have been muddied only if the memory- intact and memory-impaired animals would not have been analyzed separately as the memory strength of the memory-impaired mice is significantly lower than that of the memory-intact mice (and that of the mice at all other delays).

*Specific comments:*

*1) The sample sizes are low (n=4 per group) for these types of analyses, and may contribute to the lack of statistical clarity with respect to the CA1/CA3 dissociation.*

It is not clear which question Reviewer 2 wants us to address but it is important to point out that because of the strenuous character of the studies involving the combination of behavioral data and molecular imaging based on the detection of *Arc* expression, the standard number of animals for which *Arc* detection is performed varies between n=3-5 per group (see for example Guzowski et al., Nature Neuroscience, 1999; Vazdarjanova et al., J of Neuroscience, 2000; Guzowski et al., PNAS, 2006; Beer et al., 2013). Of note, it is important to stress that in the present study ‘even’ with this sample size, significant statistical differences were detected in 3 out of the 4 delays studied (i.e. it is possible to detect significant differences with this size of sample).

By ‘the lack of statistical clarity with respect to the CA1/CA3 dissociation’, we gather that the reviewer refers to the fact that the difference between CA1 and CA3 ‘s recruitment for the retrieval of 6 months-old memory failed to reach significance (Figure 3). This is true, however there is a clear visual trend even for this time point and CA1 and CA3’s activity differs for 3 of the 4 other delays studied. Moreover, even though the direct t-test comparison between CA1 and CA3 activity levels at 6 months failed to reach significance, comparison to 0 in each area at this time point suggested a clear difference since only CA1 was found to be engaged. Lastly, an even more stringent statistical analysis of the data using two way anova with areas and delay as factors revealed a delay by area interaction effect indicating a robust difference in the recruitment of CA1 and CA3 over time. This was stated in the original manuscript and has now be made even clearer by rephrasing the paragraph as follows:

“Importantly, this was only the case for CA3 as revealed by a significant region by delay interaction and further statistical analysis showing that, in contrast to CA3, CA1 was engaged for all delays and overall more activated than CA3 (CA1 and CA3 comparisons to 0: all ps<.033; area effect: F(1,30)=62.74, p<.001; delay effect: F(4,30)=20.15, p<.001; region by delay interaction effect: F(4,30)=5.84, p=.001; post-hoc analysis: CA1 vs CA3: all ts>3.89 and ps<.030 but for 6 months t=2.36, p=.099 and 1 week n.s; Figure 3).”

*2) It is not clear to me why freezing levels would be predicted to decline in the 1-year group* – *is there a citation the authors could include? The only paper that I am aware of-Gale et al. (2004) did not observe a drop off at super long delays.*

Reviewer 2 likely refers to the following sentence of the manuscript:

“Finally, as predicted, freezing levels of half of the one-year ‘shock’ group (n=4) were quite low (24.76 ± 2.2% in average as opposed to 62.85 ± 5.4% for the other half of the group)”.

There is indeed very little data published in animal on the topic. In humans though, important interindividual differences have been reported in the ability of retrieving memory over time (see for a review Lindenberger, Science, 2014). In essence by “as predicted’, we meant “… as predicted *by pilot experiments and suggested by studies in humans*”.

To address Reviewer 2’s comment we have now rephrased the sentence mentioned above into: “Finally, as predicted *by pilot experiments and suggested by studies in humans* (Lindenberger, Science, 2014)about half of the one-year ‘shock’ group showed poor memory performance: the freezing level of half of the one-year ‘shock’ group (n=4) was quite low…”.

Also we would also like to point out that in Gale et al., 2004, 10 times more footshocks were delivered than in our study (10 versus 1 footshock). Hence, it is likely that the memory trace generated in this study is much stronger than in our study and therefore is likely to decay much later/slower than that in our study. In the absence of more data on the subject or direct possible comparisons, this could be one reason why Gale et al., 2004 do not observe a drop off at long delay and that we do.

*3) The authors did not see a time-dependent increase in generalization. This is quite unexpected given many previous studies (e.g., Wiltgen), and is worthy of more comment and discussion.*

Time-dependent generalization (e.g. freezing to a context that is not the conditioning context) has been reported in Wiltgen’s studies and others (Winocur et al., 2007; Wiltgen et al., 2010; Wiltgen et al., 2007) when the new context is similar to the conditioning context which is not the case in our study. A time-dependent generalization to a ‘different’ new context has also been reported though but was found to depend heavily on the experimental protocol used (preexposure to the conditioning context, multiple footshocks; Wang et al., 2009; Kitamura et al., 2012; Biedenkapp and Rudy, 2007) which was not the case either in our study. Finally, as a further support to our findings, the absence of time-dependent generalization was also observed in two other studies investigating remote and very remote memory in fear conditioning tasks (delays of 16 months or 50 days, Gale et al., 2004; Anagnostaras et al., 1999, respectively). For these reasons, we did not expect a time-dependent increase in generalization and did not observe one in the present experiment.

To address Reviewer 2’s comment, we now state in the Results section the following:

“Of note, a time-dependent generalization of the freezing behavior to a similar context has been reported in some studies testing subjects 15 to 42 days after conditioning i.e. for the retrieval of early remote memory (Winocur et al., 2007; Wiltgen et al., 2010; Wiltgen et al., 2007) as well as a generalization to different contexts depending on experimental conditions (multiple footshocks, preexposure to conditioning context etc.) (Wang et al., 2009; Kitamura et al. 2012; Biedenkapp and Rudy, 2007). […] Similar findings were also observed in studies investigating the retrieval of memory at remote delays (50 days and 16 months, respectively; Anagnostaras et al., 1999; Gale et al., 2004).”

*In fact, this idea of time-dependent generalization is brought in the Discussion (second paragraph) and the authors use it to explain the absence of CA3 activation (because representation is degraded sufficiently to make pattern completion not possible). But they don't see any evidence for this in their behavior.*

As much as we understand that a lack of pattern completion could lead to generalization, this is not our interpretation of the data (one of the reasons being that indeed in our hands no generalization takes place). Hence, we rather remain cautious and restrict our speculation to the fact that the lack of CA3’s engagement might reflect a failure in pattern completing representations.

*4) Correlation analysis. I found these quite unconvincing. To support the argument that memory performance and Arc levels are related the authors need to find a correlation within the shocked group. The current analysis only really shows that shocked animals have more Arc.*

The point of our correlation analysis is indeed to show that shocked animals have more *Arc* than controls that were not shocked but were exposed to the exact same experimental conditions with the goal of giving further support to the fact that *Arc* expression and memory performance are tightly related. Significant correlations were only found at time our data suggest CA1 and CA3 are engaged in retrieving memory (i.e. not for CA3 for very remote memories). Thus, including shock and no shock group in the analysis does not automatically yield significant correlations.

Another motivation for our approach was that the variance of the imaging data (% *Arc* + cell) in the shock or the no-shock groups was quite low (in other words, activity levels for a given performance are quite similar within groups) which renders separate correlation analyses basically meaningless (Faul et al., 2007, Behavior Research Methods). Also of importance, correlation analysis is rarely performed in this type of study principally due to the low statistical power linked to a low sample size because of the strenuous character of the studies (Guzowski et al., Nature Neuroscience, 1999; Vazdarjanova et al., J Neuroscience, 2000; Guzowski et al., PNAS, 2006). However, in a recent study of the leader in the field (C. Barnes), correlation analyses between imaging results and behavioral indices of controls and experimental animals were performed in a similar manner as we did ((Hartzell et al., J. of Neurosc., 2013). For these reasons, we have not altered Figure 3 (B to E).

Reviewer #3:

*[…] Overall, I think the results are novel and that the methods are sound. So I think this will make a nice contribution to the literature. But I did have a few suggestions. 1) The dissociation between the hippocampal regions and the parahippocampal regions seems well supported by the results. However, the apparent dissociation between CA1 and CA3 is not nearly as compelling. It seems to me that both regions are involved in retrieval and they both decrease across delays, it’s just that CA3 starts off lower and so it approaches nonsignificance earlier. In order to support a claim of a dissociation here it would be necessary to show a significant interaction between regions (CA1 va CA3) and forgetting rate. As it is, it’s just that at the longer delays CA3 activity is not significantly above threshold whereas CA1 activity is. Given CA1 was always showing a larger memory effect it is perhaps not too surprising. I think the other dissociation is very interesting, so the results will be of considerable interest to many researchers, however I think that the discussion of the CA1* –

*CA3 dissociation needs to be toned down (unless the interaction is significant).*

We gather that Reviewer 3 refers to Figure 3 and an interaction between ‘regions (CA1 and CA3) and ‘delays’ as memory was intact in all groups featured in Figure 1 (i.e. there is no forgetting at stake in this study, only levels of activity during the retrieval of memory after different delays, with animals having a comparable memory strength).

As mentioned by Reviewer 3 it is of importance to find a ‘region’ by ‘delay’ interaction here to argue for a functional dissociation of CA1 and CA3. This is the case and was reported in the original manuscript in the following sentence:

“In a striking contrast, CA1 was engaged for all delays (comparison to 0: all ps<.033; Figure 3) and was overall more activated than CA3 (area effect: F(1,30)=62.74, p<.001; delay effect: F(4,30)=20.15, p<.001; *interaction effect:F(4,30)=5.84, p=.001*; post-hoc analysis: CA1 vs CA3: all ts>3.89 and ps<.030 but for 6 months t=2.36, p=.099 and 1 week n.s).”

To address the comment of Reviewer 3, we have now made this point even clearer by rephrasing the sentence as follows:

“Importantly, this was only the case for CA3 as revealed by *a significant region by delay interaction* and further statistical analysis showing that, in contrast to CA3, CA1 was engaged for all delays and overall more activated than CA3 (CA1 and CA3 comparisons to 0: all ps<.033; area effect: F(1,30)=62.74, p<.001; delay effect: F(4,30)=20.15, p<.001; *region by delay interaction effect: F(4,30)=5.84, p=.001*; post-hoc analysis: CA1 vs CA3: all ts>3.89 and ps<.030 but for 6 months t=2.36, p=.099 and 1 week n.s; Figure 3)”.

*2) I found the Introduction to be a bit confusing. For example, I did not feel like I had a good idea of what the existing literature has shown us with respect to the involvement of these MTL regions in studies of fear conditioning across these types of retention intervals.*

The Introduction and findings of the present manuscript focus on the role of the different MTL areas in the retrieval of very remote memories for which little data is available: all studies published for this time point have been cited in the Introduction of the original manuscript. References related to the role of CA1 and CA3 in more recent memories could also be found within the Nakazawa et al., 2002 and Kesner, 2007 reviews, which include but do not focus on fear conditioning studies as we did not intend to exclusively model this type of memory, but instead to use a task recruiting MTL regions for which a memory trace would persist one year after conditioning.

Nevertheless, to address Reviewer 3’s comment we have now added references that principally focus on fear conditioning, focus on more recent memories and have been published after 2007 to avoid redundancy with the reviews originally cited.

These references have been added to the following sentence of the Introduction:

“The hippocampal subfields CA1 and CA3 are functionally segregated and quite a few studies have investigated their role in recent memory, for example in the retrieval of contextual fear conditioning (Tanaka et al., 2014; Wheeler et al., 2013; Remaud et al., 2014; Daumas et al., 2005; Rampon et al., 2000; McHugh and Tonegawa, 2009; Hunsaker et al., 2009; Goshen et al., 2011; see for reviews: Nakazawa et al., 2004; Kesner, 2007).

Moreover, it was not entirely clear why the specific delays were examined in the current study.

To address Reviewer 3’s comment the following sentences have been added to the Introduction:

“Hence, the extent to which CA3 contributes to the retrieval of remote and very remote memory is not known”.

“Here, we address these issues using a contextual fear conditioning task by testing whether CA3 contributes to a comparable extent as CA1 to the retrieval of recent (1 day and 1 week- old), early remote (1 month-old) and very remote memories (6 months or 1 year-old), the latter corresponding roughly to 20 and 40 years-old memories in humans based on life expectancy, *which are among the delays the most investigated in studies focusing on very remote memory* (Figure 1).”